# Visual Three-Dimensional Reconstruction Based on Spatiotemporal Analysis Method

Xiaoliang Meng , Fuzhen Sun *, Liye Zhang , Chao Fang and Xiaoyu Wang

School of Computer Science and Technology, Shandong University of Technology, Zibo 255000, Shandong, China
* Correspondence: sunfuzhen@sdut.edu.cn

**Abstract:** To accurately reconstruct the three-dimensional (3D) surface of dynamic objects, we proposed a wrapped phase extraction method for spatiotemporal analysis based on 3D wavelet transform (WT). Our proposed method uses a 2D spatial fringe image combined with the time dimension and forms a 3D image sequence. The encoded fringe image sequence's wrapped phase information was extracted by 3D WT and complex Morlet wavelet, and we improved the wrapped phase extraction's accuracy by using the characteristics of spatiotemporal analysis and a multi-scale analysis of 3D WT, then we reconstructed the measured object by wrapped phase unwrapping and phase height transformation. Our simulation experiment results show that our proposed method can further filter the noise in the time dimension, and its accuracy is better than that of the one- (1D) and two-dimensional (2D) WT wrapped phase extraction method and the 3D Fourier transform wrapped phase extraction method because the reconstructed spherical crown's RMSE value does not exceed 0.25 and the PVE value is less than 0.95. Our results show that the proposed method can be applied to the dynamic 3D reconstruction of a real human thoracic and abdominal surface, which fluctuates slowly with respiration movement, further verifying its effectiveness.

**Keywords:** three-dimensional wavelet transform; spatiotemporal analysis; complex Morlet wavelet; wrapped phase extraction; three-dimensional reconstruction





## 1. Introduction

Three-dimensional reconstruction technology based on structured light has the advantages of non-contact, full-field, and high resolution, and is widely used in biomedical, machine vision, and other fields [1]. The wrapped phase extraction methods in structured light include the phase-shifting [2] and Fourier fringe analysis (FFA) methods [3]. For the phase-shifting method, multiple fringe images with different initial phases are obtained by shifting the phase, and the wrapped phase can be calculated by multiple fringe images. The phase-shifting method has the advantages of high resolution and accuracy; however, this method needs at least three fringe images to calculate the unwrapped phase, making it unsuitable for the 3D reconstruction of a dynamic object surface. The Fourier fringe analysis method only needs one fringe image to calculate the wrapped phase, making it suitable for the 3D reconstruction of the dynamic process. However, when Fourier transform is used to deal with this kind of fringe image, it is prone to produce large errors because of spectrum aliasing. To make up for Fourier transform's deficiencies, some time-frequency analysis techniques have been introduced into the field of 3D reconstruction, such as improved Fourier transform [4], windowed Fourier transform [5], wavelet transform [6], and S transform profilometry [7,8].

The window function used by Fourier transform is fixed; therefore, its reconstruction resolution is limited. WT's multi-resolution analysis ability, as well as the related characteristics of the mathematical operation, make WT more suitable for processing non-stationary signals because it has stronger anti-interference abilities. Phase extraction of the WT method requires the Morlet wavelet [9], which is suitable for fringe images with relatively slow

change. The Morlet wavelet has low sensitivity to noise, and its frequency-domain distribution characteristics provide a good foundation for fringe phase extraction, which has advantages for the instantaneous phase reconstruction of fringe images.

One-dimensional WT can be performed on the spatial-carrier-frequency fringe images [10], and the phase of the wavelet coefficients at the wavelet ridge can be extracted to obtain the wrapped phase. Similar to the 1D WT method, 2D WT can calculate its wrapped phase by scaling and translating in the 2D direction and extracting the phase of the wavelet coefficients at the wavelet ridge. Studies have shown that the 2D WT method's accuracy for the wrapped phase extraction of fringe images is better than that of the 1D WT method [11]. In 2D WT profilometry, the literature [12] proposed a new wavelet-ridge extraction method based on cost function, which can accurately find the optimal wavelet ridge and obtain the wrapped phase by extracting the phase at the wavelet ridge. The literature [13] used the 2D real Mexican hat wavelet function to construct 2D continuous complex wavelet. The 2D complex wavelet has a radial asymmetric frequency envelope and directional selectivity, which can improve the local edge and sub-wavelet matching ability.

As a new technology, deep learning has great potential in the field of 3D measurement, and it is increasingly being applied in the field of structured-light 3D measurement [14]. The literature [15] proposed a wrapped phase extraction method based on deep learning, in addition [16] to a 3D shape reconstruction technique based on speckle image and artificial neural networks. In this method, a single speckle pattern was used as the input to the depth CNN model, which was composed of multiple layers and nodes, and the output was the measured object's 3D point cloud information. Three different CNN models were used for comparison. The experimental results showed that the YOLOUNet model performed best. The literature [17] also proposed a 3D measurement method based on deep learning, which takes a composite fringe image as input and obtains the unwrapped (absolute) phase information through a neural network. A 3D reconstruction technology combining structured-light technology with deep convolutional neural network was also presented, which can transform 2D images into corresponding 3D depth maps without additional processing [18]. Additionally, an intelligent multi-coding deep learning technology was developed, which required only two images to obtain the unwrapped phase information of the encoded image, which has high accuracy and robustness [19].

To sum up, there are various methods for 3D reconstruction of dynamic objects, each of which has its advantages and disadvantages. Table 1 shows a specific comparison.

**Table 1.** Merits and issues of current 3D reconstruction method.

| Ref. | Method | Merits | Issues |
| --- | --- | --- | --- |
| [3–5] | Fourier transform-based method | Only need one pattern to calculate the wrapped phase | Lack of local analysis ability |
| [6,9–13] | Wavelet transform-based method | Excellent multi-resolution analysis ability | Wavelet function is not unique, and the choose wavelet function is significant |
| [14–19] | Deep learning-based method | The measurement accuracy is high | Its reliance on the training sets |

In view of the advantages of the WT [20] wrapped phase extraction method, combined with the slowly changing characteristics of human thoracic and abdominal surface with respiration, our paper proposes a structured-light 3D reconstruction method based on 3D WT. The human thoracic and abdominal surface fringe image sequence is used to carry out 3D WT, and the accuracy of wrapped phase extraction is improved by the multi-resolution and spatiotemporal analysis characteristics of 3D WT, which can further improve the accuracy of 3D reconstruction.

## 2. Methods Section

### 2.1. Principle of Wrapped Phase Extraction by 3D WT

Three-dimensional WT deals with a fringe image sequence obtained at different times (Figure 1).

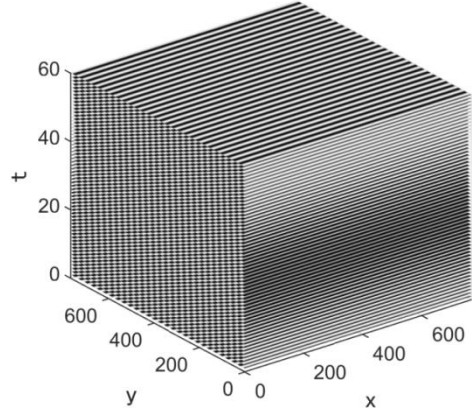

**Figure 1.** Schematic diagram of fringe image sequence.

The expression of the intensity-value function $q(x, y, t)$ is as follows:

$$q(x, y, t) = a(x, y, t) + b(x, y, t) \cos[2\pi(\alpha x + \beta y + \gamma t) + \varphi(x, y, t)] \tag{1}$$

where $a(x, y, t)$ is the background intensity; $b(x, y, t)$ is the modulation intensity; $\alpha$, $\beta$ and $\gamma$ represent the carrier frequencies of the $x$, $y$, and $t$ dimensions, respectively; $\varphi(x, y, t)$ is the phase.

The above fringe image sequence can be treated as a 3D volume, and the wavelet coefficients of the fringe image sequence after 3D WT can be expressed as follows:

$$W(\rho, \mathbf{v}, \mathbf{\eta}) = \rho^{-3/2} \int_{\mathrm{R}^3} d^3 \xi \psi^* (\rho^{-1} r\_\eta^{-1}(\xi - \mathbf{v})) q(\xi) \tag{2}$$

where $\psi(\xi)$ is the 3D wavelet function, * denotes complex conjugate operation, $\rho$ is the scale factor, $\mathbf{v}$ represents the translation vector $\mathbf{v} = (v_x, v_y, v_t)$, $\mathbf{\eta}$ contains information about three Euler angles, and $r\_\eta$ is the rotation matrix. Its parameters can be set by three Euler angles $\xi = (x, y, t)$.

The complex Morlet wavelet is suitable for the fringe image with relatively slow change. This kind of directional wavelet has slow sensitivity to noise, and its distribution characteristics in the frequency domain provide a good function for fringe phase extraction. It has advantages in the fringe image's instantaneous phase reconstruction. Therefore, we used the following complex Morlet wavelet function in this paper:

$$\psi_M(\xi) = \exp(i\kappa_0 \xi) \cdot \exp\left(\frac{1}{2} |\mathrm{E}\xi|^2\right) \tag{3}$$

where $\kappa_0 = (w_0, 0, 0)$, $\mathrm{E} = diag[\varepsilon_1^{-1/2}, \varepsilon_2^{-1/2}, 1]$, $\varepsilon_1 \geq 1$, $\varepsilon_2 \geq 1$.

Similar to the 2D WT method, we ignored the influence of the angle and rotation matrix, and calculated the wavelet coefficients' amplitude using the 3D WT method:

$$W_{am}(\rho, \mathbf{v}, \mathbf{\eta}) = \sqrt{\{\mathrm{Re}[W(\rho, \mathbf{v}, \mathbf{\eta})]\}^2 + \{\mathrm{Im}[W(\rho, \mathbf{v}, \mathbf{\eta})]\}^2} \tag{4}$$

where $\mathrm{Re}[\cdot]$ and $\mathrm{Im}[\cdot]$ represent the real and imaginary parts, respectively. Because the 3D WT method uses the information of the coordinate axis, scale factor, and amplitude to form five-dimensional information, its wavelet coefficients' amplitude can no longer be visually displayed in the form of graphs, such as in the 1D and 2D WT methods. When the

scale factor is $\rho_r$, if the wavelet coefficients' amplitude obtained by the 3D WT method is the largest, this position is called the 3D WT wavelet ridge.

According to the wavelet coefficients $W(\rho_r, \mathbf{v}, \mathbf{\eta})$ at the wavelet ridge after 3D WT, the expression of the fringe image sequence's wrapped phase $\varphi(\mathbf{v})$ can be calculated as follows:

$$\varphi(\mathbf{v}) = \tan^{-1}\left\{ \frac{\mathrm{Im}[W(\rho_r, \mathbf{v}, \mathbf{\eta})]}{\mathrm{Re}[W(\rho_r, \mathbf{v}, \mathbf{\eta})]} \right\} \tag{5}$$

According to the above analysis, the 3D WT calculation process is as follows:

1. Obtain the signal $I(x, y, t)$, and set the initial value of the scale factor $\rho_{\min}$, then select the 3D Morlet wavelet function $\psi_M(x, y, t)$ for calculation.
2. Shift the wavelet function to three space directions (the distances are $v_x, v_y$, and $v_t$, respectively); then, we can obtain the sub-wavelet function. Repeat the previous step until the end of the whole signal.
3. Change the value of the scale factor $\rho$ and repeat the above two steps until the maximum scale factor $\rho_{\max}$ is calculated.
4. When the scale factor of the wavelet coefficients is $\rho_r$, if the amplitude of the wavelet coefficients at this position is the maximum, this position is called the wavelet ridge. According to the wavelet coefficients $W(\rho_r, \mathbf{v}, \mathbf{\eta})$ at the wavelet ridge, the fringe image sequence's wrapped phase information can be obtained.

### 2.2. The Relation between Scale Factor and Periodic Length of Fringe Image

Although we used the 3D WT method to process the fringe image sequence, the projected fringe pattern used in this paper is one-dimensional (Figure 2).

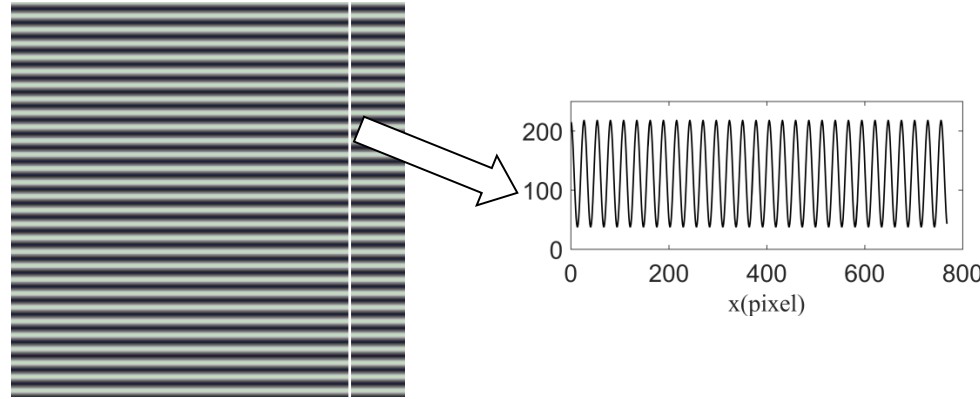

**Figure 2.** Fringe pattern and intensity value of one column.

By deducing the relation between the scale factor and the first dimension's periodic length, the relation between the scale factor and the second and third dimension's periodic lengths can also be obtained. Therefore, we take the 1D signal as an example. Without considering the influence of background factors, the expression of the 1D cosine function can be written as follows:

$$q(x) = b\cos(w_1 x + \varphi) \tag{6}$$

where $w_1 = 2\pi/T_1$ is the periodic length and $\varphi$ represents the phase.

The expression after the Fourier transform of Equation (6) can be expressed as follows:

$$\hat{q}(w) = \pi b[\delta(w - w_1)e^{i\varphi} + \delta(w + w_1)e^{-i\varphi}] \tag{7}$$

According to the WT power spectrum, we derived the relation between the wavelet scale factor and the periodic length. The expression of the WT power spectrum is as follows:

$$
\begin{aligned}
P_W(\rho, \nu_x) &= \frac{1}{\Gamma^2} \frac{|W(\rho,\nu_x)|^2}{\rho} \\
&= \frac{1}{\Gamma^2} \left| \frac{1}{2\pi} \int_{-\infty}^{+\infty} \hat{q}(w)[\hat{\psi}(\rho w)]^* e^{iw\nu_x} dw \right|^2 \\
&= \frac{1}{\Gamma^2} \left| \frac{1}{\sqrt{2\pi}} \int_{-\infty}^{+\infty} \hat{q}(w) e^{\frac{-(\rho w - w_0)^2}{2}} e^{iw\nu_x} dw \right|^2
\end{aligned}
\tag{8}
$$

where $\Gamma$ is a constant. Substituting Equation (7) into (8), we have:

$$
\begin{aligned}
P_W(\rho, \nu_x) &= \frac{1}{\Gamma^2} \left| \frac{1}{\sqrt{2\pi}} \int_{-\infty}^{+\infty} \hat{q}(w) e^{\frac{-(\rho w - w_0)^2}{2}} e^{iw\nu_x} dw \right|^2 \\
&= \frac{1}{\Gamma^2} \left| \frac{1}{\sqrt{2\pi}} \int_{-\infty}^{+\infty} [\pi b \delta(w - w_1) e^{i\varphi} + \pi b \delta(w + w_1) e^{-i\varphi}] e^{\frac{-(\rho w - w_0)^2}{2}} e^{iw\nu_x} dw \right|^2 \\
&= \frac{\pi b^2}{2\Gamma^2} \left| e^{\frac{-(\rho w_1 - w_0)^2}{2}} e^{i(w_1 \nu_x + \varphi)} + e^{\frac{-(\rho w_1 + w_0)^2}{2}} e^{-i(w_1 \nu_x + \varphi)} \right|^2
\end{aligned}
\tag{9}
$$

where $\rho > 0$ and $w_0 > 5$ for the wrapped phase extraction of fringe images, we usually select $w_0 = 6$, then:

$$
e^{\frac{-(\rho w_1 + w_0)^2}{2}} < e^{-\frac{w_0^2}{2}} \approx 1.5 \times 10^{-8}
\tag{10}
$$

and ignoring the above item, we have:

$$
P_W(\rho, \nu_x) = \frac{\pi b^2}{2\Gamma^2} e^{-(\rho w_1 - w_0)^2}
\tag{11}
$$

Because the position with the largest power corresponds to the largest amplitude of the wavelet coefficients, this position is the wavelet ridge. To obtain the maximum value of $P_W(\rho, \nu_x)$, we take the derivative of the above formula, and then we have:

$$
\frac{\partial P_W(\rho, \nu_x)}{\rho} = (\rho w_1 - w_0) w_1 e^{-(\rho w_1 - w_0)^2} = 0
\tag{12}
$$

The unique solution of the above equation can be calculated by $\rho = w_0/w_1 = w_0 T_1/(2\pi)$.

For the fringe image with a given periodic length, we can theoretically obtain the optimal value of the scale factor, and the scale factor will fluctuate around this value in our actual calculations. Therefore, according to the above formula, the calculated scale factor value can be used as the intermediate value to set an appropriate value range in the actual calculation process, which can further reduce the computation time and improve the efficiency of wrapped phase extraction.

## 3. Experiment and Discussion

### 3.1. Simulation Experiment

To evaluate our proposed method's surface reconstruction error, we used hemispheres with gradually increasing radii for our 3D reconstruction simulation experiment. The hemisphere surface formula is as follows:

$$
h(x, y, t) = \sqrt{r^2(t) - [(X(x, y) - 384)^2 + (Y(x, y) - 384)^2]}
\tag{13}
$$

where the range of $X(x, y)$ and $Y(x, y)$ are [1, 768]. The expression of the hemisphere radius changing with time is $r(t) = 200 + t$, where $t$ is an integer and $t \in [1, 60]$.

We projected the fringe pattern with a periodic length of 27 pixels onto the gradually increasing hemispherical surface. The fringe image sequence was composed of deformed fringe images at different times, which we calculated using the 1D, 2D, and 3D WT methods;

then, we carried out the phase unwrapping and phase height conversion process to obtain the hemispheres' 3D reconstruction results at different times. We selected the t = 10, t = 30, and t = 50 3D reconstruction results for comparison, and the results are shown in Figure 3. The figure shows that these three methods can reconstruct the 3D shape of the measured hemispherical surface; however, the reconstruction error cannot be judged directly.

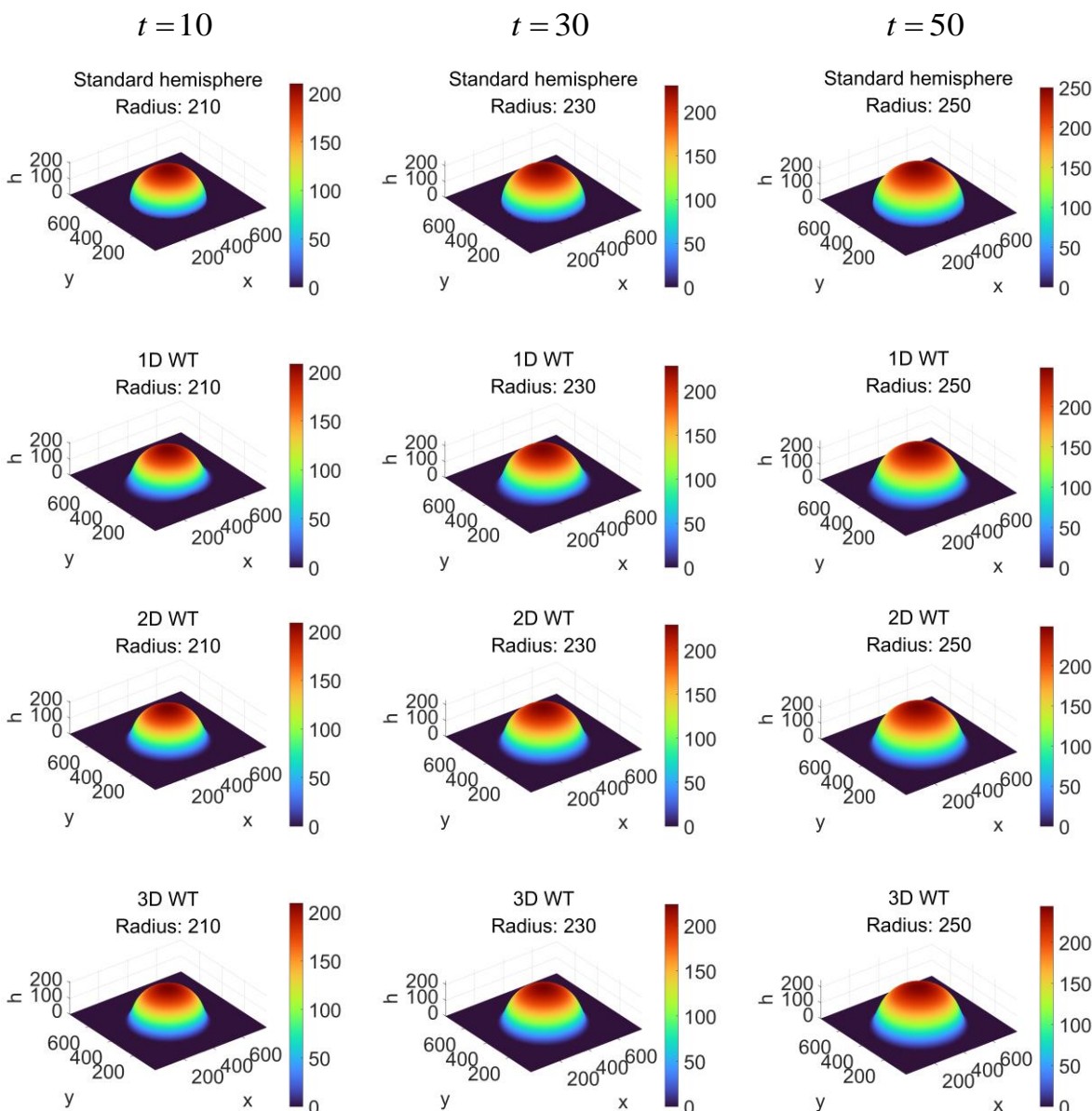

**Figure 3.** Standard hemisphere surface and hemisphere 3D reconstruction result with different methods.

To quantitatively evaluate the surface reconstruction errors of these three methods, we intercepted the hemisphere surface using the cross-section of h = 150 and compared the reconstruction errors of each truncated spherical crown surface with the standard spherical crown surface (Figure 4). The figure shows that the root mean square error (RMSE) and peak valley error (PVE) of the spherical crown surface obtained by our proposed method are smaller than those obtained by the 1D and 2D WT methods, indicating that the proposed method has higher accuracy.

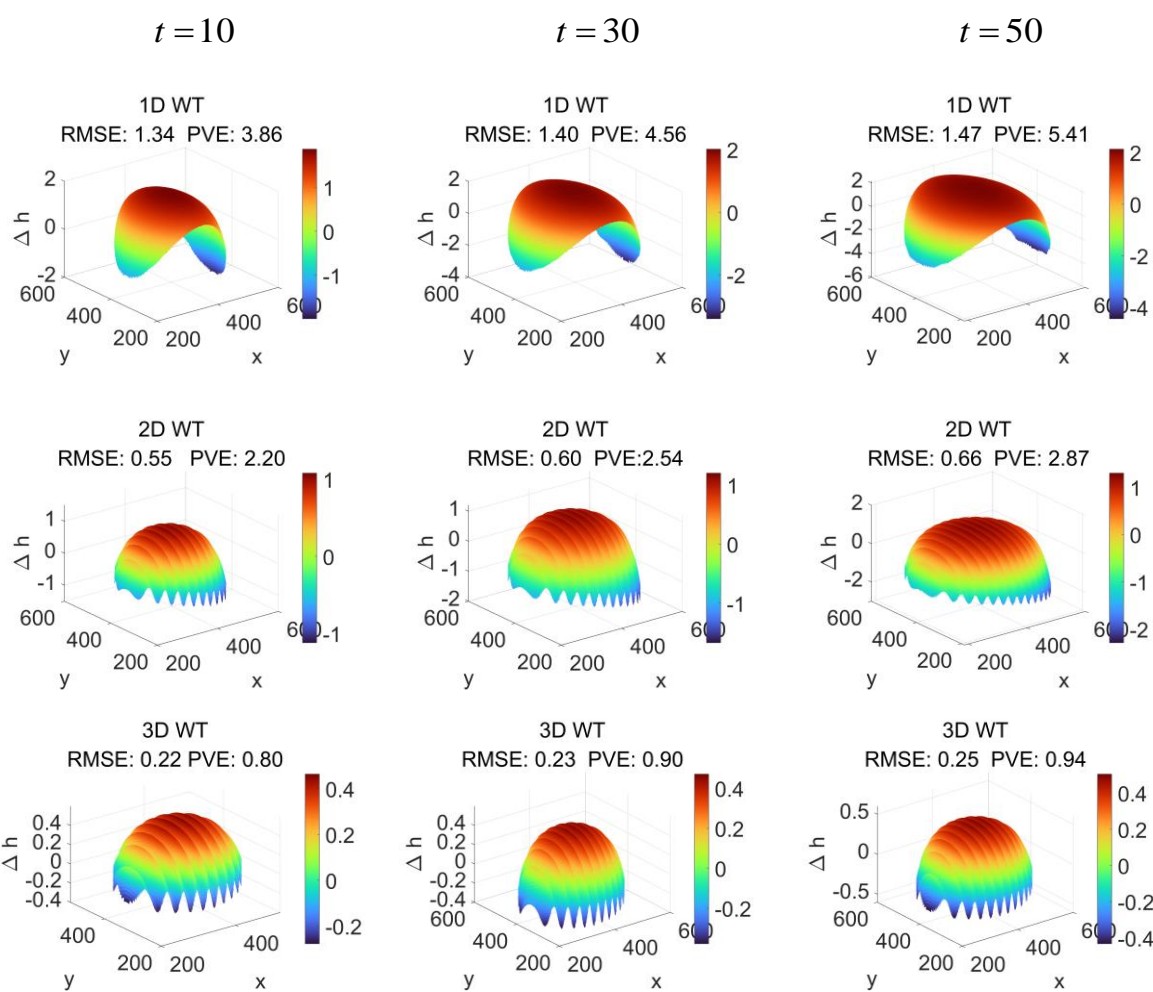

**Figure 4.** Spherical crown reconstruction error with different method.

To compare our method with the 3D FFA method in the literature [21], we also used the gradually increasing hemispherical surface and the t = 10, t = 30, and t = 50 3D reconstruction results. In Table 2, we added the peak signal-to-ratio (PSNR) with [22] to compare the spherical crown surface errors of these two methods.

**Table 2.** Comparison between the 3D FFA method and our method.

|        |      | t = 10 | t = 30 | t = 50 |
|--------|------|--------|--------|--------|
|        | RMSE | 0.30   | 0.26   | 0.27   |
| 3D FFA | PVE  | 1.56   | 1.59   | 1.67   |
|        | PSNR | 58.59  | 59.83  | 59.50  |
|        | RMSE | 0.22   | 0.23   | 0.25   |
| 3D WT  | PVE  | 0.80   | 0.90   | 0.94   |
|        | PSNR | 61.28  | 60.90  | 60.17  |

Table 2 shows that the RMSE and PVE of the 3D WT method are less than those of the 3D FFA method, indicating that the 3D WT method has higher accuracy in the reconstruction of a curved surface, which further verifies the advantages of our proposed method.

### 3.2. Human Thoracic and Abdominal Surface 3D Reconstruction Experiment

3.2.1. Projected Pattern Design

The fringe patterns in the red, green, and blue channels with different periodic lengths are generated by computer, and these three channels can combine into a composite color

fringe pattern, which can be used to project onto the human thoracic and abdominal surface. The camera captures dynamic fringe images at fixed time intervals and obtains deformed color fringe image sequences. In our experiment, the fixed projected pattern reduced the conversion and establishment of the projected pattern, with only one composite color fringe pattern projected, which can reduce the image-acquisition time and lay a foundation for the realization of dynamic 3D reconstruction. The intensity value of the red, green, and blue channels can be expressed as follows:

$$g_r(x,y) = a(x,y) + b(x,y)\cos(2\pi f_r x + \phi_r(x,y)) \tag{14}$$

$$g_g(x,y) = a(x,y) + b(x,y)\cos(2\pi f_g x + \phi_g(x,y)) \tag{15}$$

$$g_b(x,y) = a(x,y) + b(x,y)\cos(2\pi f_b x + \phi_b(x,y)) \tag{16}$$

where $a(x,y)$ is the background intensity; $b(x,y)$ represents the fringe contrast; $f_r$, $f_g$, and $f_b$ are the carrier frequencies of the red, green, and blue channels, respectively; $\phi_r(x,y)$, $\phi_g(x,y)$, and $\phi_b(x,y)$ represent the phase. Figure 5 shows composite fringe pattern.

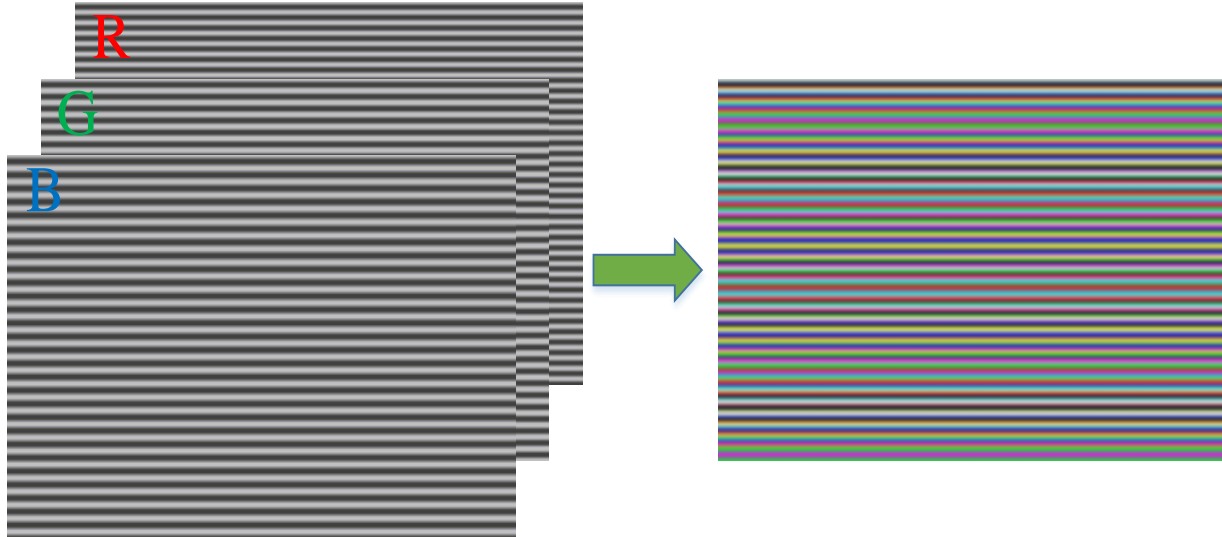

**Figure 5.** Composition of color-encoded fringe pattern.

### 3.2.2. Three-Dimensional Reconstruction System and Color Separation

Our paper uses the red, green, and blue channels of color images to encode the fringe pattern. We used a 1624 × 1236 resolution 3CCD industrial camera (AT-200 GE model type). The DLP projector can project high-resolution images, and, considering the requirements of projection resolution and color restoration degree, we selected the INFocus IN82 as the device to project the fringe pattern. The fringe pattern carrier frequencies are 1/21, 1/24, and 1/27, and the projected fringe pattern resolution is 1024 × 768. Figure 6 shows a schematic diagram of the 3D reconstruction system.

We selected 60 frames of fringe images for our experiment. Figure 7 shows the measured area truncated from the image sequence of the human thoracic and abdominal surface. Figure 7a shows the surface after the projection of the composite color fringe pattern. We truncated the image region of the human thoracic and abdominal surface for color-coupling correction and separation [23]. The separated fringe image sequences are shown in Figure 7b–d.

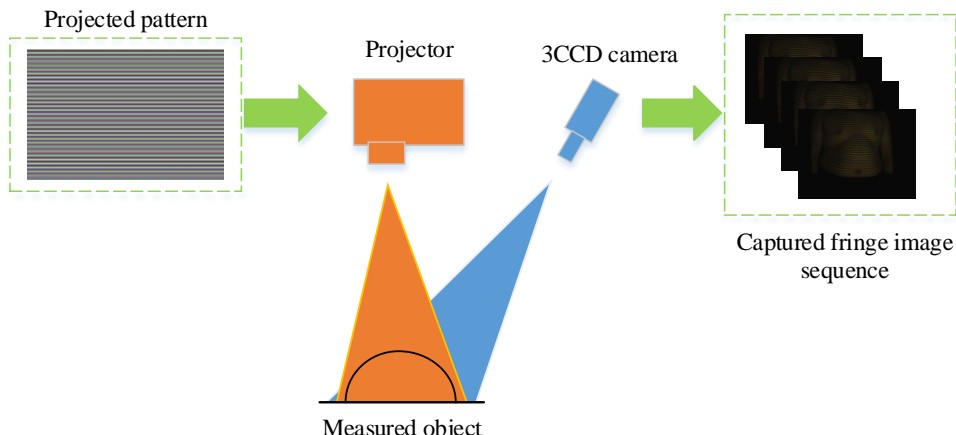

**Figure 6.** Schematic diagram of 3D reconstruction system.

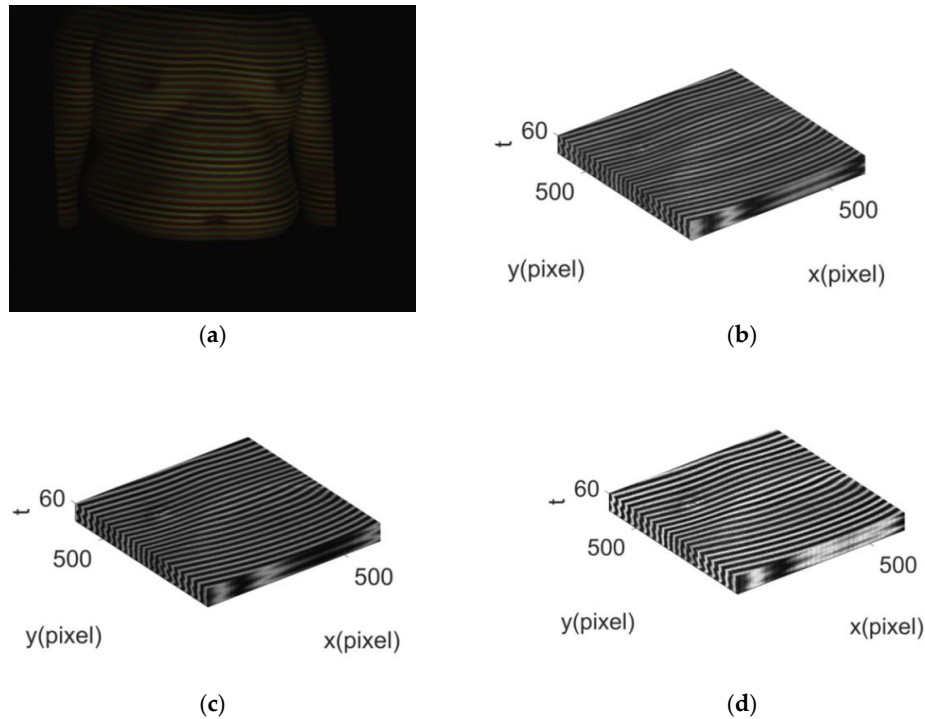

**Figure 7.** Real human thoracic and abdominal surface reconstruction. (**a**) Thoracic and abdominal surface after projecting color fringe pattern; (**b**) separated fringe image sequence of red channel; (**c**) separated fringe image sequence of green channel; (**d**) separated fringe image sequence of blue channel.

### 3.2.3. Wrapped Phase Unwrapping

Phase unwrapping [24] can be divided into two approaches: spatial-domain and time-domain phase unwrapping. Spatial-domain phase unwrapping can be realized through the spatial coding of at least one image and has the advantages of the fast acquisition of encoded images but the disadvantages of low resolution and poor robustness. Time-domain phase unwrapping requires multiple images to be encoded in the time domain, and has the advantages of high resolution and strong robustness but the disadvantages of long time-acquisition time. We adopted the fixed spatial encoding of a composite color fringe pattern, which only needs to capture one deformed fringe image, obtained three different images with different carrier frequencies through color separation, and finally used the time-domain phase unwrapping method to realize the phase unwrapping. Our method not only makes use of the advantages of the fast acquisition of encoded images by spatial-

domain encoding, but also the advantages of high resolution and the strong robustness of time-domain phase unwrapping.

The time-domain phase unwrapping method includes three phase unwrapping strategies, namely, the multi-frequency [25], heterodyne [26], and number-theoretical methods [27]. The heterodyne method is the most widely used, the multi-frequency method has the lowest anti-interference ability, and the number-theoretical method is the most complex. Additionally, the heterodyne method has the highest probability of large errors in the phase resolution near the projected pattern's edge. Therefore, we constructed a three-frequency temporal phase unwrapping model based on the geometric relation between the wrapped phase of different frequency fringe images and their unwrapped phases. The phase unwrapping method based on this model improves the operation speed of phase unwrapping and reduces the probability of large errors.

### 3.2.4. Three-Dimensional Reconstruction of Human Thoracic and Abdominal Surface

To compare the reconstruction results of our 3D WT method with the 3D FFA method in the literature [16], we selected and compared the 3D-reconstructed results of the human thoracic and abdominal surface at three different times for these two methods (Figure 8).

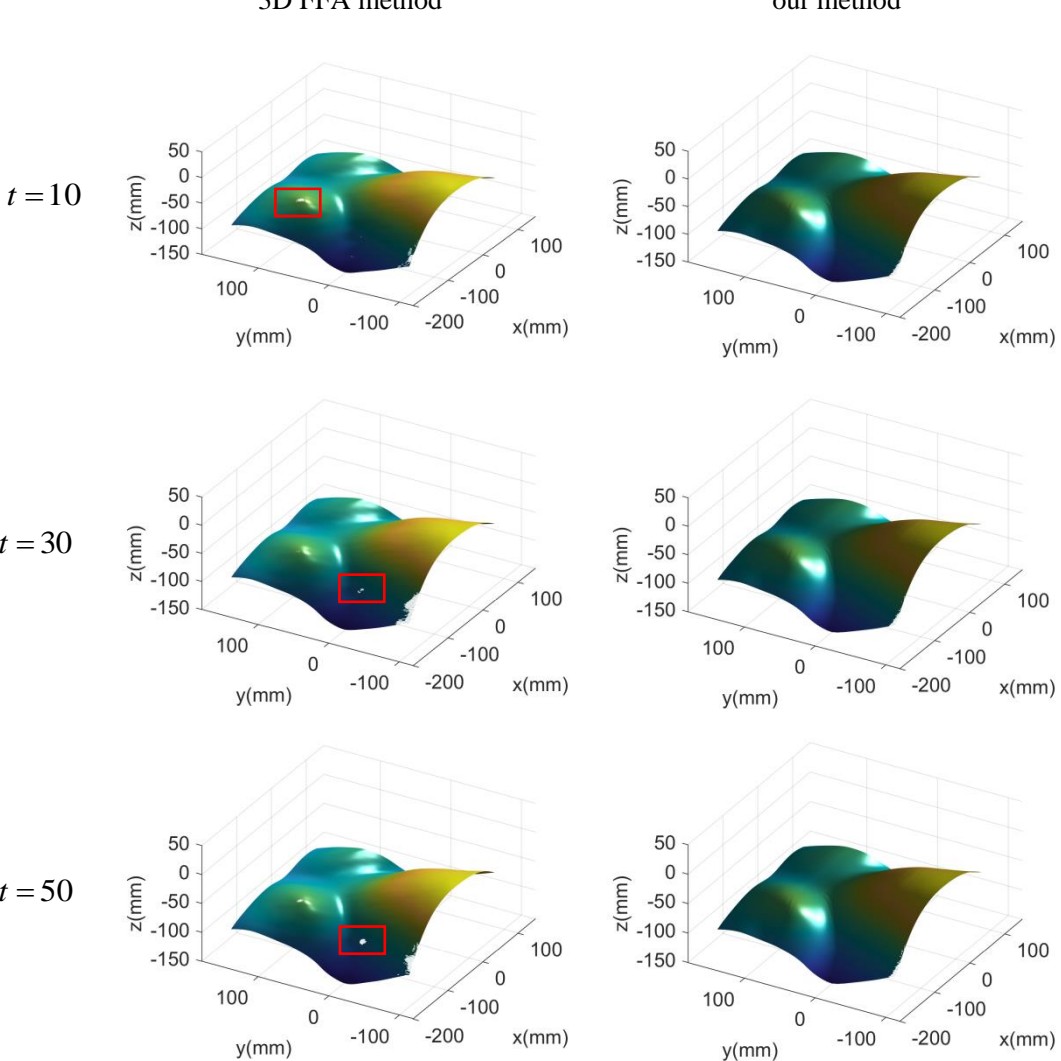

**Figure 8.** Thoracic and abdominal surface 3D reconstruction result of three different times with different method.

Figure 8 shows that both the 3D WT and 3D FFA method can correctly reconstruct the 3D shape of a human thoracic and abdominal surface; however, our method's reconstructed surface was finer and smoother. This is because the wavelet transform method has the ability of local- and multi-resolution analysis, increasing its effect-of-detail performance. While the Fourier fringe analysis (FFA) method only has the ability of global analysis, when the spectrum of the useful and interference signal is aliased, the wrapped phase error increases, thus increasing the measurement error. Therefore, the 3D FFA method eliminates some small areas of the reconstructed human thoracic and abdominal surface due to large errors, resulting in some small areas of surface blank (red box in Figure 8), which further demonstrates the effectiveness and advantages of our 3D WT method.

When the 3D WT method is used for human thoracic and abdominal surface 3D reconstruction, the selection of the scale-factor interval has a certain influence on the 3D reconstruction result. Here, we used an interval of 1 with 0.1 of scale factors to show the 3D reconstruction result's details (Figure 9). When the scale-factor interval changes from 1 to 0.1, our proposed method can obtain smoother reconstruction results; however, the computation time increases accordingly.

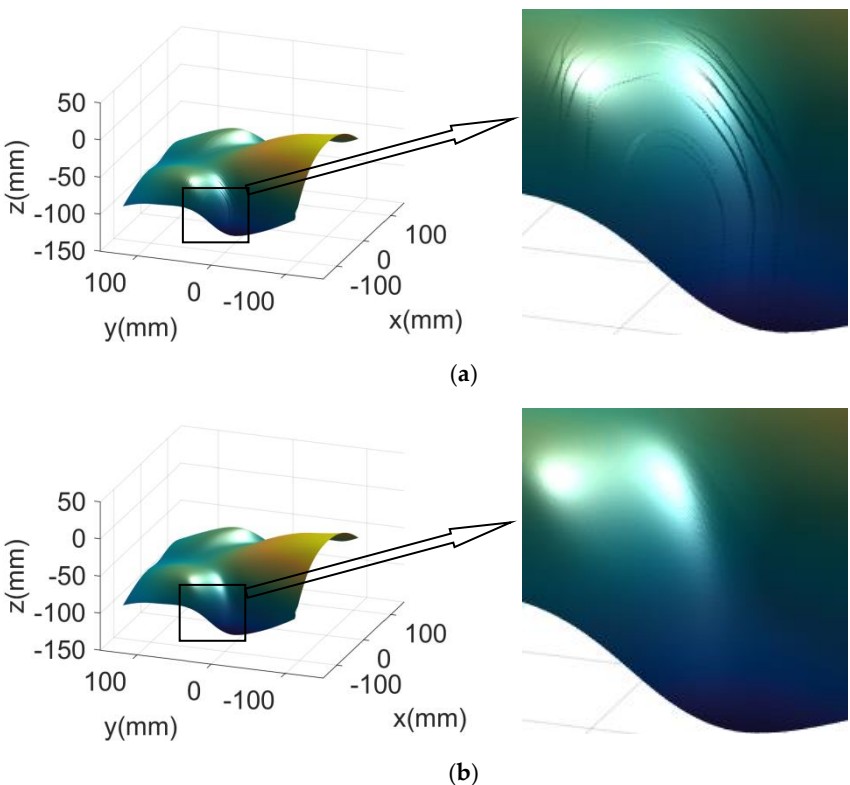

**Figure 9.** Three-dimensional reconstruction results using 3D WT method with different scale-factor intervals. (**a**) Scale-factor interval is 1; (**b**) Scale-factor interval is 0.1.

## 4. Conclusions

The 3D reconstruction of a dynamically changing object is a challenging problem in the field of visual measurement. We make use of the dynamic changing object's movement in the time-dimension direction, which we combine with the 2D spatial image to form a 3D image sequence. We used 3D WT method to conduct spatiotemporal analysis, and we carried out the hemisphere simulation experiment to compare and analyze the accuracy of our proposed spatiotemporal analysis method. Our simulation results show that our proposed method's accuracy is not only better than that of the 1D WT and 2D WT wrapped phase extraction methods, but also the 3D FFA method. Our proposed method's effectiveness was further verified by the 3D reconstruction of a real human thoracic and abdominal surface.

There are several aspects of our proposed approach that need further improvement. First, from the perspective of engineering applications, subsequent research needs to consider using parallel computing to further improve the 3D reconstruction algorithm's speed. Second, because the deep learning method has been widely used in the field of 3D reconstruction, its adoption should be considered in the future to achieve an accurate 3D reconstruction of the measured object.

**Author Contributions:** X.M. conceived and designed the experiments; F.S. analyzed the data; L.Z., C.F. and X.W. wrote the paper. All authors have read and agreed to the published version of the manuscript.

**Funding:** This research was funded by the National Natural Science Foundation of China (grant number 62001272).

**Data Availability Statement:** The authors confirm that the data supporting the findings of this study are available within the article.

**Conflicts of Interest:** The authors declare no conflict of interest.

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
