# Peer review of "Visual Three-Dimensional Reconstruction Based on Spatiotemporal Analysis Method"

_electronics, doi:10.3390/electronics12030535_

Round 1

Reviewer 1 Report

In this paper, authors have suggested a visual three-dimensional reconstruction Based on the spatiotemporal analysis method. The paper is written well and can be a good contribution. However, I would suggest authors to revise and resubmit. 

1. The issue and objective of the work should be outlined/highlighted in the abstract.

2. The abstract should highlight some results at the end.

3. The literature review should be further improved with an in-depth discussion of the existing works.

4. I would suggest the authors to refer some more recent works to compare them and present them in a table (with their merits and issues).

5. Since the work is based on three-dimensional reconstruction/reversibility, I would suggest the authors include one more measure/parameter known as peak signal-to-noise ratio. 

6. This PSNR will find the image/object disparity/similarity after reversibility, authors can go through the following works to find further details on PSNR:

https://doi.org/10.1016/j.optlaseng.2020.106245

https://doi.org/10.1007/s10044-022-01104-0

7. Also, it should give a clear idea of what are statistical and real-time parameters are obtained.

8. The result section can be improved by mentioning the future direction and implications. 

Reviewer 2 Report

1. The cited references in this paper are very limited and almost completely ignore the progress of deep learning in this field, which is also fully reflected in the follow-up experiments. Furthermore, there are too few reviews on 3D WT-based methods, which lead to the raising of thesis questions and insufficient investigation of existing methods.

2. The experiments in this paper are very limited, and there is also a lack of sufficient comparative experiments with commercial algorithms or SOTA research methods in this field. In particular, the 3D FFA methods are only compared in simulation experiments. This is far from enough to verify the effectiveness and superiority of the algorithm in this paper. Experiments should be greatly increased, whether it is the comparison with the SOTA method or the diversity of experimental data.

Reviewer 3 Report

Paper is well presented, few recommendations are listed below:

Authors should add pixel value and specify the significance of the work carried out in the paper in comparison to existing literature in the abstract.

The future scope and potential application of the work should be explored in the conclusion.

Round 2

Reviewer 1 Report

The authors have revised the manuscript as per my suggestion. 

Reviewer 2 Report

Nothing